# Inhibition of DNA Repair by Inappropriate Activation of ATM, PARP, and DNA-PK with the Drug Agonist AsiDNA

**DOI:** 10.3390/cells11142149

**Published:** 2022-07-08

**Authors:** Nathalie Berthault, Ptissam Bergam, Floriane Pereira, Pierre-Marie Girard, Marie Dutreix

**Affiliations:** 1Institut Curie, PSL Research University, CNRS, INSERM, UMR 3347, 91405 Orsay, France; nathalie.berthault@curie.fr (N.B.); floriane.pereira@curie.fr (F.P.); pierre-marie.girard@curie.fr (P.-M.G.); 2Université Paris-Saclay, CNRS, UMR 3347, 91405 Orsay, France; 3Institut Curie, PSL Research University, CNRS, INSERM, UMS 2016, Multimodal Imaging Centre, 91405 Orsay, France; ptissam.bergam@curie.fr; 4Université Paris-Saclay, CNRS, UMS 2016, 91405 Orsay, France

**Keywords:** DNA damage signaling, double-strand breaks repair, DNA repair inhibitor, AsiDNA

## Abstract

AsiDNA is a DNA repair inhibitor mimicking DNA double-strand breaks (DSB) that was designed to disorganize DSB repair pathways to sensitize tumors to DNA damaging therapies such as radiotherapy and chemotherapy. We used the property of AsiDNA of triggering artificial DNA damage signaling to examine the activation of DSB repair pathways and to study the main steps of inhibition of DNA repair foci after irradiation. We show that, upon AsiDNA cellular uptake, cytoplasmic ATM and PARP are rapidly activated (within one hour) even in the absence of irradiation. ATM activation by AsiDNA leads to its transient autophosphorylation and sequestration in the cytoplasm, preventing the formation of ATM nuclear foci on irradiation-induced damage. In contrast, the activation of PARP did not seem to alter its ability to form DNA repair foci, but prevented 53BP1 and XRCC4 recruitment at the damage sites. In the nucleus, AsiDNA is essentially associated with DNA-PK, which triggers its activation leading to phosphorylation of H2AX all over chromatin. This pan-nuclear phosphorylation of H2AX correlates with the massive inhibition, at damage sites induced by irradiation, of the recruitment of repair enzymes involved in DSB repair by homologous recombination and nonhomologous end joining. These results highlight the interest in a new generation of DNA repair inhibitors targeting DNA damage signaling.

## 1. Introduction

DNA damage can have deleterious effects, as it interferes with DNA replication and transcription, and it can ultimately result in mutations and chromosomal aberrations, and cell death in highly proliferating cells. Therefore, the induction of damage by genotoxic agents or irradiation is commonly used to kill cancer cells in the clinic. DNA damage-signaling and repair machineries have been developed in all living systems to counteract the adverse consequences of DNA lesions, to preserve genome integrity, and to ensure its faithful transmission [1,2]. These mechanisms play an important role in the cancer’s resistance to treatments.

Among all the damage induced by radiation in the cell, DNA double-strand breaks (DSBs) are the most deleterious. To protect cells against such events, several complementary repair pathways exist: nonhomologous end joining (NHEJ); its alternative form, PARP-dependent (alt-NHEJ); and homologous recombination (HR), all operating on different end substrates and involving a complex cascade of proteins recruitment at the damage sites. NHEJ and alt-NHEJ directly mediate the re-ligation of the broken DNA molecule and are active in all phases of the cell cycle, while HR directs repair via the use of a homologous DNA sequence as a template, and is primarily active in only the S/G2 phase of the cell cycle [3,4,5]. Not only is the choice of DSBs repair pathway cell-cycle-dependent, but it is also determined by the recruitment of the first proteins binding the DNA ends [6,7]. Noticeably, the three pathways share some proteins at the early steps of the repair process and use specific enzyme complexes at later stages, thus rendering the precise scenario of strand break repair extremely difficult to apprehend. For example, PARP1 promotes the binding of MRE11 and NBS1 [8], which form the MRN complex with RAD50, essential for HR. In contrast, PARP1 competes with Ku proteins, which form with DNA-PKcs the Ku protein-DNA-dependent protein kinase system (DNA-PK) involved in the repair of DNA DSBs by NHEJ [9,10]. The kinetics of recruitment at damage sites is also different. PARP and Ku reach DNA breaks in around 1 s post-damage, whereas the recruitment of MRN takes at least 10 s [11].

ATM is a kinase that belongs, together with DNA-PKcs, to the PI3-like kinases group [9]. The recruitment and activation of ATM and DNA-PKcs follow the kinetics of MRN and Ku recruitment, respectively. Therefore, DNA-PKcs is recruited about 10 s before ATM. Both kinases can phosphorylate chromatin-bound histone variant H2AX (*γ*−H2AX) in response to DSBs [12,13]; this phosphorylation is observed around 1 min after damage, a time scale that is considerably slower than that for PARP1 recruitment. Within such a short time, it is difficult to decipher what the exact events triggering the further cascade of repair protein recruitment are. To dissociate the activation of PARP and DNA-PK from the events occurring at the damage sites, we used AsiDNA (also called coDbait or DT01), which are short DNA molecules mimicking DSBs [14] that activate PARP [15] and DNA-PK [16], even in the absence of DNA damage on chromosomes. In this study, we show that AsiDNA also triggers ATM activation, and that the kinetics of ATM, DNA-PK, and PARP activation is different, ranging from a few minutes (ATM and PARP) to a few hours (DNA-PK). Using irradiation, we induced damage at different times after the start of AsiDNA treatment, key times corresponding to full activation of ATM, PARP, DNA-PK, or chromatin modification by AsiDNA. This method allowed for us to discriminate events that precede chromatin modification from others, and determine the respective roles of kinases and poly-ADP-polymerase in DNA repair.

## 2. Materials and Methods

### 2.1. Cell Lines and Treatments

Cell cultures were performed with the following human cell lines: MDA-MB-231 breast cancer cell line (ATCC, HTB26), SK-MEL-28 melanoma cell line (from Institut Curie), MRC5VI SV40-transformed normal fibroblasts (ATCC, CCL-171), AT5BIVA SV40-transformed ATM-deficient fibroblasts [17], Hela cells stably silenced (PARP1^KD^) or not (Ctr) for PARP-1 [18], M059K DNA-PK proficient glioblastoma cell line (ATCC, CRL-2365), and M059J DNA-PK deficient glioblastoma cell line (ATCC, CRL-2366). Cells were grown according to the supplier’s instructions and maintained at 37 °C under a humidified atmosphere of 5% CO_2_ except for MDA-MB-231 cells, which grow without CO_2_. Before use, all cell lines were verified by short tandem repeat profiling (Geneprint 10, Promega) at 10 different loci (TH01, D21S11, D5S818, D13S317, D7S820, D16S539, CSF1PO, AMEL, vWA, TPOX) and tested negative for mycoplasma contamination with LookOut^®^ Mycoplasma PCR Detection Kit (Sigma Aldrich Chimie S.a.r.l, Saint-Quentin-Fallavier, France).

AsiDNA is a 64-nucleotide (nt) oligodeoxyribonucleotide consisting of two 32 nt strands of complementary sequence connected through a 1.19bis (phospho)-8-hydraza-2-hydroxy-4-oxa-9-oxo-nonadecane linker with cholesterol at the 5′-end and three phosphorothioate internucleotide linkages at each of the 5′ and the 3′ ends (Agilent, USA). The sequence is: 5′-X GsCsTs GTG CCC ACA ACC CAG CAA ACA AGC CTA GA L-CLTCT AGG CTT GTT TGC TGG GTT GTG GGC AC sAsGsC-3′, where L is an amino linker, X a cholesteryl tetraethylene glycol, CL a carboxylic (hydroxyundecanoic) acid linker, and s is a phosphorothioate linkage. The stock concentration of AsiDNA dissolved in water was at 40 mg/mL, and AsiDNA was directly added to the cell culture medium to the final concentration of 150 µg/mL (time zero), and incubated at 37 °C for the indicated times.

PARP inhibitor AZD2281 (olaparib) (Cliniscience, #orb154674), DNA-PK inhibitor NU7026 (Sigma, #N1537), ATM inhibitor KU55933 (Selleckchem, #S1092), and ATR inhibitor Ceralasertib (Selleckchem, #AZD6738) were used at the final concentrations of 1, 25, 10, and 0.5 µM, respectively.

### 2.2. Electronic Microscopy

For TEM immunogold labeling, SK28 cells were treated with 150 µg/mL of biotinylated AsiDNA for different times. Then, the cells were fixed with 2% paraformaldehyde (PFA), 0.2% glutaraldehyde in 0.1 M phosphate buffer pH 7.4, washed with phosphate-buffered saline (PBS 1×), embedded in 10% (*wt*/*vol*) gelatin, and 2.3 M sucrose. Mounted gelatin blocks were frozen in liquid nitrogen, and ultrathin sections were achieved with an ultracryomicrotome (UC6 Leica). Ultrathin cryosections of about 90 nm thickness were collected with 2% (*vol*/*vol*) methylcellulose, 2.3 M sucrose, labeled with 15 nm gold beads coated with streptavidin. The contrast was enhanced with a mixture of methylcellulose–uranyl acetate (pH 4) on ice for 10 min before observation under the electron microscope. Samples were examined on a JEOL 2200FS energy-filtered (20 eV) field-emission gun electron microscope operating at 200 kV using a 2048 × 2048 pixels ssCCD camera (Gatan, U.S.). The distribution of AsiDNA molecules in cell compartments was obtained by counting the number of gold particles in pictures taken randomly per cell compartment defined by morphology. Around 100 fields were analyzed.

### 2.3. Antibodies, Irradiation-Induced DNA Damage, and Immunostaining

For immunostaining, cells were grown on glass coverslips, treated, and stopped when they were 80% confluent. Irradiation was performed using ^137^Cs gamma rays (GSR D1, Gamma Service, U.K.) at a 0.5 Gy/min dose rate. Cells were grown at 37 °C and 5% CO_2_ (except for MDA-MB-231) for the indicated times, and fixed for 20 min in 4% PFA in PBS 1×, permeabilized in 0.5% Triton X-100 in PBS 1× for 10 min, blocked with 2% bovine serum albumin (BSA) in PBS 1× (blocking buffer) for 1 h, and incubated successively with each primary antibody in blocking buffer for 1 h at room temperature (RT). All secondary antibodies were used at a dilution of 1/200 in blocking buffer for 30 min at RT, and DNA was stained for 10 min with 4′,6-diamidino-2phenylindole (DAPI). The coverslips were kept on slide with glycergel mounting medium (Dako,# C0563). The following primary antibodies were used: mouse anti-phospho-histone H2A.X (Ser139) clone JBW301 at 1/500e (Millipore, #05-636), polyclonal rabbit anti-phospho-histone H2A.X (Ser139) at 1/100e (Abcam, #ab11174), rabbit anti-phospho-HSP90a at 1/100e (Biogenes GmbH, Germany), rabbit polyclonal anti-53BP1 at 1/200e (Cell signaling Technology, #4937), rabbit polyclonal anti-Rad51 at 1/1000e (Calbiochem, #PC130), mouse anti-BRCA1 (D9) at 1/100e (Santa Cruz Biotechnology, #sc-6954), mouse monoclonal anti-phospho-ATM Ser1981 at 1/100e (Millipore, #05-740). The primary rabbit polyclonal antibody anti-NBS1 at 1/100e (Novus Biological, #NB100-143) was used after a wash of cells for 30 s with cytoskeletal (CSK) buffer containing 0.5% Triton X-100 just before fixation. The following secondary antibodies were used: goat antimouse or goat antirabbit IgG (H + L) conjugated with Alexa Fluor 488 or Alexa Fluor 633 (Molecular Probes, ThermoFisher).

Imaging was performed using a Leica SP5 confocal system attached to a DMI6000 stand using a 63/1.4 objective. A minimum of 100 nuclei in at least 10 different fields distributed randomly on each sample were imaged. Each assay was at least duplicated. Images were processed using a macro written to automate the foci recognition into ImageJ software complemented with the LOCI bioformat plugin (http://www.loci.wisc.edu/ome/formats.html, accessed on 15 December 2021). The focus quantification was statistically analyzed using the Mann–Whitney test with StatEl software (ad Science, Orsay, France) implemented in Excel.

### 2.4. Inducing Photodamage with Laser

Cells were seeded on glass coverslips and transfected with an XRCC1-pEYFP plasmid [19] (a gift from P. Radiccella, IRCM, CEA-DSV, Fontenay-aux-Roses, France), pEGFP-C3-hPARP-1 plasmid [20] (a gift from V. Schreiber, UMR7175 CNRS, ESBS, Illkirch, France) or pEGFP-C1-FLAG-XRCC4 [21] (Addgene plasmid, # 46959). Plasmid transfection was performed 48 h before the addition of AsiDNA or protein inhibitors using the jetPRIME transfection reagent (PolyPlus, #114-15). The cells were maintained alive in complete phenol red-free medium inside a metal chamber under a controlled temperature of 37 °C. A Leica SP5 confocal system coupled to a multiphoton system (Coherent Chameleon Ultra) was used to generate localized DNA damage into the nucleus. A predamage sequence of two images was acquired within a period of 2–3 s. To mainly generate double- or single-strand breaks, we used the multiphoton laser at 810 nm (set to 30% output for 500 ms) and 405 nm, respectively, and focused onto a spot of 176 nm diameter. For XRCC4 recruitment, the cells were pretreated with Hoechst 33342. The laser was set to a maximal output of 100 ms for XRCC1 recruitment, and 500 ms for PARP1 recruitment. Records were produced using the appropriate argon laser line (514 nm for YFP or 488 nm for GFP) at 2 s intervals for the following 132 s.

The recruitment quantification was obtained using ImageJ software and a macro to automate data extraction by calculating the mean photodamage spot intensity relative to the mean nucleus intensity, excluding the spot corrected by the background and photobleaching. The maximum of the recruitment was estimated with the kinetics. Quantifications were statistically analyzed using the Mann–Whitney test with the StatEl software (ad Science, France) implemented in Excel.

### 2.5. ELISA Assay for PARylation

A sandwich ELISA was used to detect poly(ADP-ribose) (PAR) polymers in AsiDNA-treated cells using plates coated with mouse anti-PAR, and further revealed with rabbit anti-PAR (see details in Appendix A). For nuclear and cytoplasmic extractions, the REAP method was used [22] using an ice-cold solution of 20 mM Tris HCl pH 7.5, 1 mM EDTA, 1 mM EGTA, 20 mM sodium pyrophosphate (Na_4_P_2_O_7_), 25 mM sodium fluoride, 1 mM β-glycerophosphate, 1 mM orthovanadate (Na_3_VO_4_), 1 μg/mL leupeptin, 1 mM PMSF, and 0.1% IGEPAL (NP40).

## 3. Results and Discussion

### 3.1. Small DNA Fragments AsiDNA Activate DNA-PK, PARP, and ATM with Different Kinetics

The kinetics of AsiDNA cellular uptake and cellular distribution was monitored by electronic microscopy, and revealed that the molecules are preferentially located in the cytoplasm and reach maximal concentration at 5 h, with most of the molecules in the cytoplasm and only one-tenth in the nucleus (Figure 1A,B). We also investigated AsiDNA uptake and its stability over time by flow cytometry using fluorescent AsiDNA molecules (Cy5.5-AsiDNA). Nearly 98% of the cells could take up AsiDNA molecules at 1 h post incubation. If the maximum of molecules present at the cellular level were at 5 h (Appendix AA), approximately half of them were still detected at 120 h (Appendix AB). Then, we monitored the activation of DNA-PK and ATM kinases, and PARP polymerase (collectively called “damage-signaling enzymes”) upon AsiDNA treatment by analyzing the modifications to proteins that are specific of their activities. Therefore, DNA-PK activity was estimated by the phosphorylation of HSP90α and H2AX, events which are strictly DNA-PK dependent after AsiDNA treatment (Appendix A) [14,23], ATM activation was monitored by its autophosphorylation at serine 1981 [24], and PARP activity was estimated by measuring the poly-ADP-ribose (PAR) content in the cells [25]. The kinetics of activation of the three DNA damage-signaling enzymes by AsiDNA were all different (Figure 1C; Appendix A), and did not reflect the uptake kinetics. PARP and ATM were activated within a few minutes, whereas DNA-PK activation took several hours. Moreover, the persistence of the activation strongly varied between enzymes. ATM shows the shorter activation time with a full return to basal state within 24 h, while DNA-PK activation decreased slowly after 24 h, and the one of PARP was still persistent 5 days after adding AsiDNA. The specific kinetics of each DNA damage-signaling enzyme was observed in the three cell lines used in this study (Figure 1C). The activations of each enzyme seemed to be independent of each other, as the inactivation or inhibition of one enzyme did not change the ability of AsiDNA to induce the activity of the others (Figure 2). Next, we further characterized the respective role of the different DNA damage-signaling events and their consequences.

### 3.2. ATM Activation and Trapping in the Cytoplasm

ATM activation is associated with a shuttling of the protein from the cytoplasm to the nucleus [26]. The immunostaining of AsiDNA-treated cells revealed that cytoplasmic ATM is phosphorylated at Ser1981 (P-ATM) 1 h after adding AsiDNA in the medium, and remains excluded from the nucleus during the first 5 h (Figure 3A,C). These results should be seen in relation to those described above showing that the maximal activation of ATM was reached at 1 h after adding AsiDNA (Figure 1C), which is itself detected in the cytoplasm but not in the nucleus (Figure 1B).

AsiDNA treatment prevents the recruitment of P-ATM in nuclear repair foci after irradiation for at least 24 h. Though cytoplasmic ATM phosphorylation is strongly reduced from 5 h of AsiDNA treatment (Figure 1C, Appendix A), the formation of radiation-induced P-ATM foci remains impaired for more than 24 h, suggesting that either ATM is not available in the nucleus or/and it cannot be activated at irradiation-induced damage site after AsiDNA uptake (Figure 3A). A functional MRE11/RAD50/NBS1 (MRN) complex is required for ATM foci formation after irradiation [27], so we tested the ability of NBS1 to form radiation-induced foci in AsiDNA-treated cells. NBS1 recruitment after irradiation was not impaired 5 h after AsiDNA treatment (Figure 3D), indicating that the inhibition of P-ATM foci observed at this time point is not due to a defect of the recruitment of MRN complex at the damage site. In contrast, at 24 h, when ATM phosphorylation is undetectable (i.e., back to basal level), the formation of NBS1 and ATM foci at irradiation-induced damage sites are both impaired (Figure 3). Our results suggest that AsiDNA treatment does not prevent NBS1 recruitment at an early time (5 h), but could participate in its inhibition at 24 h by preventing the stabilization of NBS1 by ATM at the damage sites [28]. The more likely explanation is that ATM is trapped by AsiDNA directly or indirectly in the cytoplasm and cannot shuttle to the nucleus.

### 3.3. PARP Activation Properties

PARP showed a rapid activation, like ATM, with a 50-fold increase of the total cellular content of PAR already observed after 1 h of AsiDNA treatment. The PAR signal was mainly located in the cytoplasm at all time points with a PAR concentration that was 10-fold less abundant in the nucleus than that in the cytoplasm (Figure 4A). In contrast to ATM phosphorylation, the PAR level remained stable for at least 5 days (Figure 4B). To determine if the PAR persisting signal was due to the persistent activation of PARP, we added the PARP inhibitor olaparib (AZD2281) at different times after AsiDNA treatment. The inhibition of PARP activity immediately reduced the PAR amount at any time, confirming that PAR persistence was due to the continuous activation of PARP by AsiDNA, and not due to inhibition of its degradation (Figure 4B).

PARP recruitment at damaged sites is a very rapid process [8]. We used laser-induced damage to monitor the recruitment of the PARP1-EGFP enzyme in cells treated by AsiDNA. The activation of ATM and DNA-PK or PARP itself by AsiDNA did not seem to affect the capacity of PARP to be recruited at the damage sites (Figure 5A). This result was confirmed by the efficient recruitment at any time of XRCC1 (Figure 5B), which interacts with PARP [29] to require the PARP1 activity to form foci at the damage sites [30].

### 3.4. DNA-PK Activation Properties

We investigated the activation of DNA-PK through the phosphorylation of two targets: H2AX and HSP90α. Although they are common targets of ATR, ATM, and DNA-PK, both are phosphorylated by DNA-PK after AsiDNA treatment, and both show the same kinetics of phosphorylation (Figure 2C, Appendix A). All nuclei with a pan-nuclear localization of *γ*H2AX display the phosphorylation of HSP90α. However, although HSP90α has both cytoplasmic and nuclear localization, P-HSP90α was mainly detected in the nuclei at any time (Appendix A). The radiation-induced *γ*H2AX foci are currently used as markers of DSB formation after irradiation. At early times (1–5 h), when ATM was trapped in the cytoplasm (Figure 3A), and DNA-PK was poorly activated (Figure 1B), the irradiation triggered the formation of *γ*H2AX foci in the nucleus of mock-treated cells as that of cells treated with AsiDNA (Figure 3A). At a later time (24 h), when *γ*H2AX showed a pan-nuclear distribution in AsiDNA-treated cells, the irradiation did not promote its recruitment in foci (Figure 3A, Appendix A). We observed a few γH2AX foci in some cells, in addition to the pan-nuclear phosphorylation (Appendix A). Considering that all cells are capable of taking up AsiDNA at 24 h, but at various levels (Appendix A), we hypothesized that the concentration or distribution of intracellular AsiDNA in those cells is not optimal to trigger efficient DNA-PK-dependent pan-nuclear *γ*H2AX staining, and thus does not prevent IR-induced *γ*H2AX foci. We also observed that the phosphorylation of H2AX and HSP90α started to slowly decrease at 24 h after the beginning of AsiDNA treatment (Appendix A).

The initial step in DSB repair mediated by NHEJ is the rapid binding of the Ku heterodimer to the DSBs. The Ku70/80 crystal structure shows that the two subunits dimerize through the central domain to form a ring capable of accommodating two turns of double-stranded DNA (approximately 14 base pairs). Moreover, Ku binds double-stranded DNA ends with high affinity, including 5′-3′ or 3′-5′ overhangs and blunt ends [31]. At DSBs, Ku heterodimer recruits DNA-PKcs to form the active complex. Hence, the AsiDNA molecule, despite the presence of cholesterol at the 5′ end, is able to load the DNA-PK complex.

The activity and function of DNA-PKcs in end ligation or NHEJ are tightly regulated by phosphorylation modification [32,33,34]. S2056 and T2609 are two prominent autophosphorylation sites of DNA-PKcs, which are crucial for its activity in DNA repair [33]. S2056 is phosphorylated in trans, at least in vitro, in response to IR, and this phosphorylation event influences the pathway choice during the NHEJ reaction [33]. T2609 is part of the ABCDE cluster. If the ABCDE cluster does not undergo phosphorylation, DNA-PKcs remains bound to DNA ends in a Ku-dependent manner, and the NHEJ reaction does not proceed any further [33]. We found that AsiDNA treatment in the absence of irradiation did not induce the autophosphorylation of DNA-PK at S2056 and T2609 (Figure 6A), suggesting that DNA-PK complex loading onto AsiDNA is impaired in its ability to carry out DNA end ligation and DNA end resection. This agrees with the concept that AsiDNA is a decoy mimicking a double-strand break that cannot be sealed [35]. However, AsiDNA did not prevent the autophosphorylation of DNA-PK at S2056 and T2609 after irradiation, even after 24 h of AsiDNA treatment (Figure 6A), a time point at which the majority of cells displayed the pan-nuclear staining of *γ*H2AX (Figure 3A,B, Appendix A), and reduced ability of P-DNA-PK to form foci (Figure 6B). These results suggest that, despite the autophosphorylation of DNA-PK in response to IR, the repair of DSBs in AsiDNA-treated cells is impaired. This is in line with a recent publication showing that AsiDNA decreases the frequency of telomere fusions, a process that relies on classical and alternative nonhomologous end-joining repair pathways [36].

### 3.5. Consequences of the Activation of AsiDNA-Dependent Signaling on the Recruitment of Repair Proteins

We identified four statuses of DNA damage-signaling according to the time after AsiDNA treatment and the activation state of ATM, DNA-PK, and PARP (Figure 7): at 0 h, no activation; at 1 h, ATM was activated; at 5 h, PARP was activated; at 24 h, PARP and DNA-PK were activated. Therefore, we monitored the recruitment of key enzymes involved in DNA breaks repair at irradiation-induced damage sites (Table 1). For that purpose, cells were incubated with AsiDNA and exposed to irradiation at 1, 5, and 24 h post AsiDNA treatment. At 1 h, ATM, 53BP1, and XRCC4 foci were reduced (Table 1, Appendix A). At 5 h, 53BP1, XRCC4 (Appendix A), and ATM (Figure 3A,C) focus formation was inhibited. At 24 h, all the tested enzymes involved in homologous recombination (NBS1, BRCA1, RAD51) (Figure 3B,D, Appendix A) and nonhomologous end joining (P-DNA-PK, 53BP1, XRCC4) (Figure 6B; Appendix A) did not form foci after irradiation. To determine the role of ATM, DNA-PK, and PARP activation in these recruitment inhibitions, we used specific inhibitors or cells deficient in one of these proteins. We showed that ATM does not play any role in the inhibition of this focus formation at any time point, pointing to PARP and DNA-PK as the main regulators. PARP activation did not seem to play an important role at any time except in the inhibition of 53BP1 and XRCC4 foci at 1 h and 5 h (Appendix A; Table 1). At these early time points, AsiDNA reduces radiation-induced 53BP1 and XRCC4 focus formation, foci that are restored in cells inactivated for PARP (Appendix A). Luijsterburg et al. showed that PARP1 regulates the assembly of NHEJ complexes at broken chromosomes to promote efficient DNA repair [37]. Since, at 1 and 5 h of AsiDNA treatment (Figure 5), PARP1 is recruited at radiation-induced damage (Figure 5), but fails to recruit XRCC4 (Appendix A), our results suggest that the inappropriate activation of PARP by AsiDNA impaired downstream events involved in PARP-dependent DSBs repair signaling [37]. As PARP1 competes with Ku proteins at DSBs [6,10], we hypothesized that in PARP deficient cells, and at early time points (1 and 5 h), the classical Ku-dependent NHEJ (cNHEJ), which is not yet impaired by AsiDNA (Figure 6), is operating. Indeed, DNA-PK activation was mainly responsible for the inhibition of all enzyme recruitments at late time points, but had no action at earlier times (Appendix A). As these inhibitions follow the kinetics of pan-nuclear phosphorylation of the histone H2AX, part or all these inhibitions upon AsiDNA treatment could occur via the modification of the chromatin.

Collectively, these data suggest that, upon AsiDNA treatment, part of the nuclear DNA-PK complex is loaded onto AsiDNA, resulting in the activation of DNA-PKcs kinase activity, and subsequently to the uniform widespread nuclear phosphorylation of histone H2AX. We propose that the inappropriate modification of the chromatin impaired the repair of ionizing radiation-induced DNA double-strand breaks.

## 4. Discussion

Using the short DNA molecules’ AsiDNA, we demonstrated that, while ATM and PARP can detect and signal DNA ends in the cytoplasm, DNA-PK is only activated in the nucleus. The cascade of repair enzyme recruitment is highly dependent on the local modification of H2AX and its pan-nuclear phosphorylation by DNA-PK. Overactivating the DNA damage-signaling using AsiDNA molecules before the induction of damage prevents the recruitment of the repair enzymes and their organization in foci at damage sites, thereby impairing the repair of DNA double-strand breaks. The recruitment of the 53BP1 enzyme seems to be a complex mechanism that is inhibited by PARP activation in the cytoplasm at early times and by DNA-PK, probably via H2AX phosphorylation, at later times. The disorganization of the damage signaling and the formation of the repair foci may account for the defect of repair observed in AsiDNA-treated cells and tumors, and contribute to its property of sensitizing cancer cells to many DNA damaging treatments [14,38,39,40,41].

## Figures and Tables

**Figure 1 cells-11-02149-f001:**
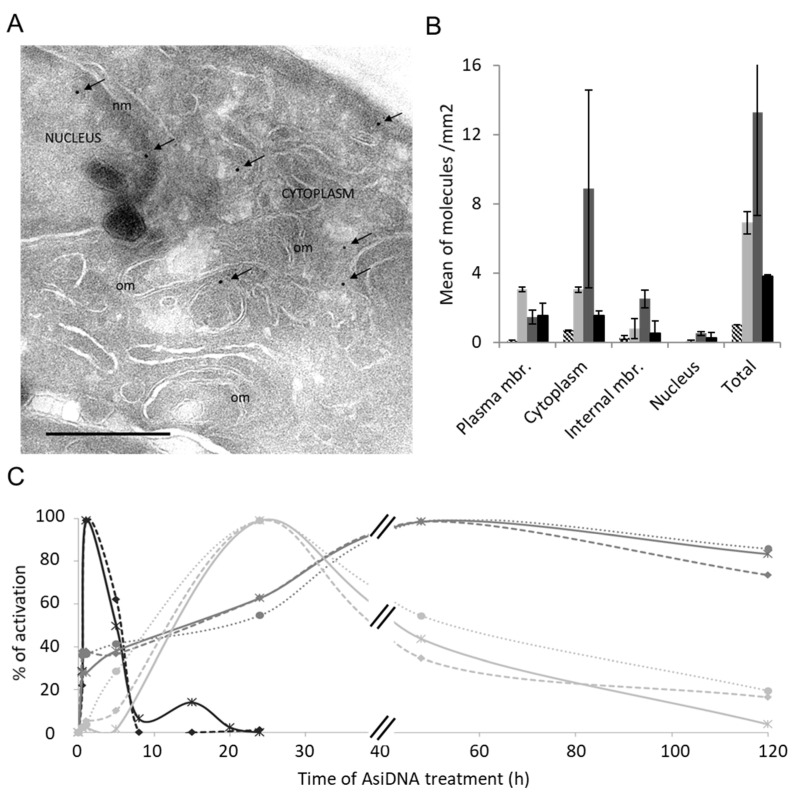
Kinetics of AsiDNA uptake and ATM, PARP1, and DNA-PK activation. (**A**) SK28 cells were treated with biotinylated AsiDNA and analyzed at different times with electronic microscopy. Typical picture after 5 h of AsiDNA treatment. Arrows indicate gold-streptavidin beads bound to biotinylated AsiDNA. om, organelle membrane; nm, nucleic membrane. Scale bar: 500 nm. (**B**) Quantification of AsiDNA distribution in the plasma membrane, cytoplasm, internal membranes, and nucleus. Cells were untreated (dashed), or treated for 1 h (light gray), 5 h (gray), or 24 h (black) with biotinylated AsiDNA before imaging. The detected signal in the untreated samples was not due to biotinylated AsiDNA but likely due to endogenous biotin. (**C**) Enzyme activation by AsiDNA in three cell lines: SK28 (continuous lines), MRC-5sv (dashed lines), and MDA-MB-231 (dotted lines). The kinetics of ATM autophosphorylation (black) was measured by immunofluorescence quantification on microscopy images. DNA-PK kinase-dependent H2AX phosphorylation (light gray) was measured by Western blot, and PARP1 activation was estimated by PAR quantification with an ELISA test (gray). Relative activation is reported as compared to maximal signal after different times of AsiDNA treatment. Symbols indicate measured data on smooth graphs.

**Figure 2 cells-11-02149-f002:**
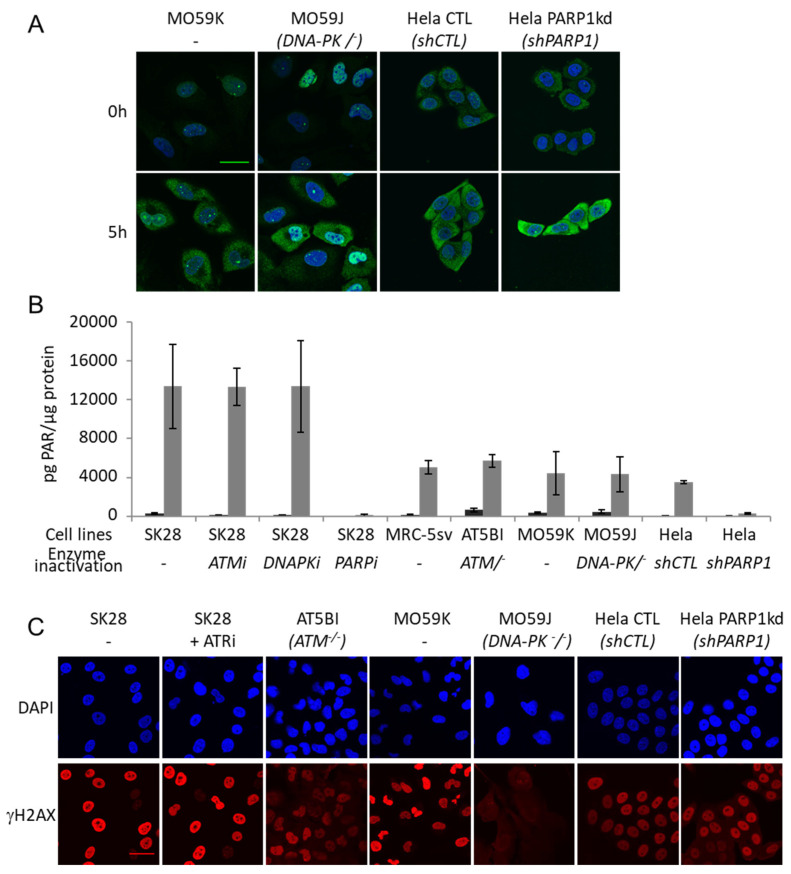
Main DNA damage-signaling enzymes are activated by AsiDNA independently of each other. (**A**) P-ATM (green) immunostaining after 5 h of AsiDNA treatment or no treatment in DNA-PK and PARP1-deficient cells; DAPI staining (blue). Scale bar: 30 µm (**B**) Quantification of PARylation by ELISA after 24 h of AsiDNA treatment (gray) or no treatment (black) in SK28 treated with ATM inhibitor (KU55933), DNA-PK inhibitor (NU7026), PARP inhibitor (olaparib) or in ATM, DNA-PK or PARP1 deficient cell lines. (**C**) Immunostaining of *γ*H2AX after 24 h of AsiDNA treatment in SK28 inhibited for ATR with ceralasertib, in ATM and DNA-PK-deficient cell lines, or in PARP1 silenced cell line. Scale bar: 30 µm.

**Figure 3 cells-11-02149-f003:**
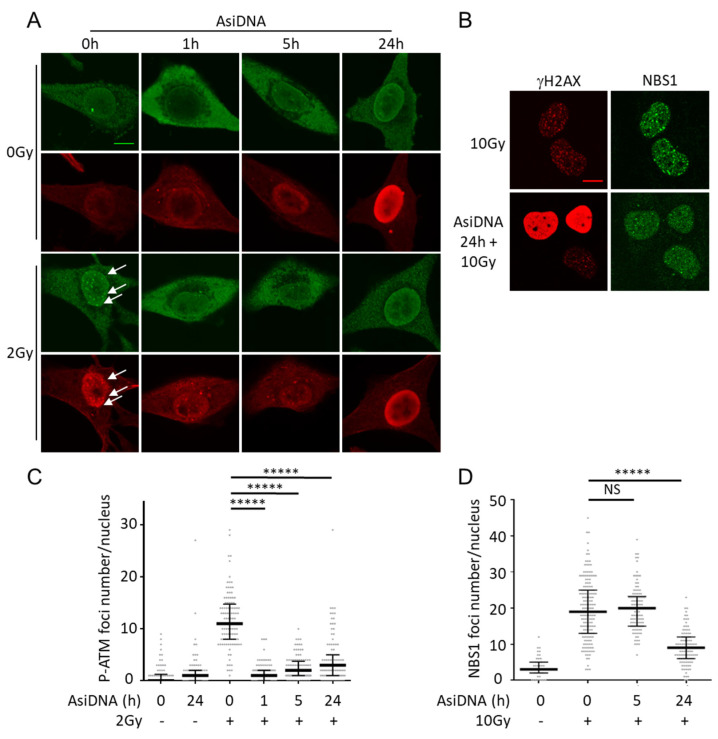
ATM phosphorylation in SK28 cells treated for different times with AsiDNA, and inhibition of NBS1 foci formation after irradiation. (**A**) ATM phosphorylation (green) and *γ*H2AX (red) were detected by immunostaining in SK28 cells treated with AsiDNA at different times indicated in abscissa and irradiated or not at 2Gy. White arrows indicate foci with colocalization of P-ATM and *γ*H2AX. Scale bar: 10 µm. (**B**) Inhibition of NBS1 recruitment after irradiation in MRC-5sv cells treated 24 h with AsiDNA and exposed to 10Gy. Scale bar: 10 µm. (**C**) Quantification of P-ATM foci per nucleus after various treatments. (**D**) Quantification of NBS1 foci per nucleus after various treatments. Mann-Whitney test: ***** *p* value < 0.00001, NS: not significant.

**Figure 4 cells-11-02149-f004:**
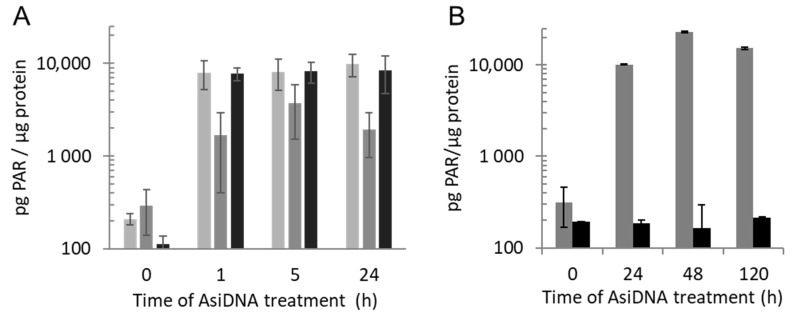
PARP activation by AsiDNA. (**A**) Quantification of PARylation in total (light gray), nuclear (dark gray) or cytoplasmic (black) extracts of SK28 was performed after different times of AsiDNA treatment. (**B**) Kinetics of PARylation in cells treated for different times with AsiDNA and challenge with (black) or without (gray) olaparib during 1 h before PAR quantification.

**Figure 5 cells-11-02149-f005:**
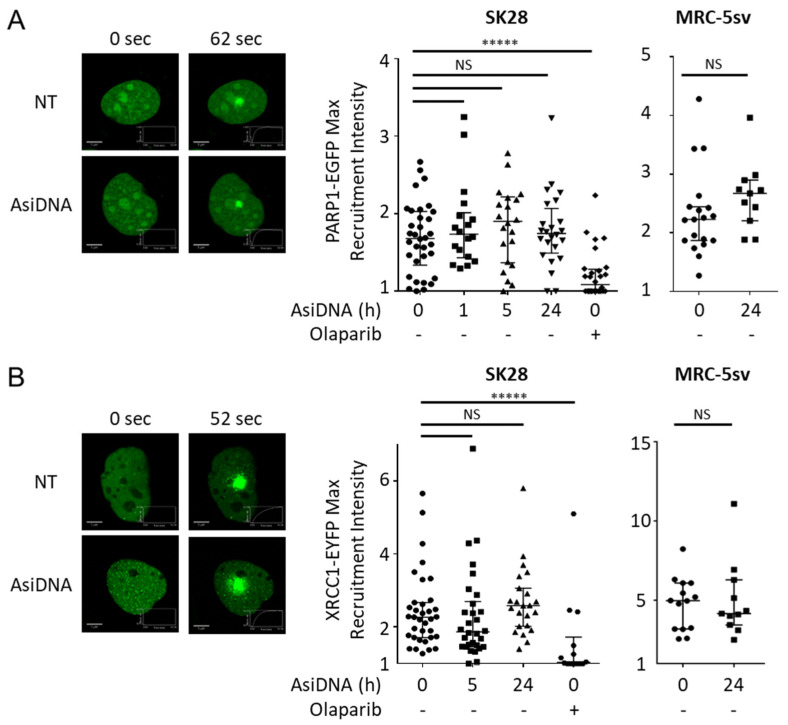
Recruitment of PARP1 and XRCC1 at laser-induced damage. SK28 or MRC-5sv cells expressing (**A**) PARP1-EGFP or (**B**) XRCC1-EYFP plasmids were treated for different times with AsiDNA or with PARP inhibitor olaparib. DNA photodamage was induced by a laser at 405 nm, and the recruitment of fluorescent proteins was monitored by video microscopy. (left) Typical pictures at the maximal recruitment time. (right) Quantification of the maximal intensity of recruitment after photodamage corrected by photobleaching and background. Mann–Whitney test: ***** *p* value < 0.00001, NS: not significant. Scale bar: 5 µm.

**Figure 6 cells-11-02149-f006:**
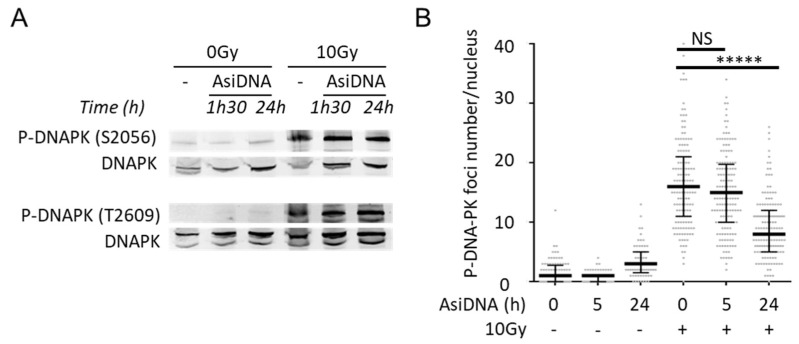
DNA-PK autophosphorylation and inhibition of its recruitment by AsiDNA. (**A**) Autophosphorylation of DNA-PK at S-2056 and T-2609 in MRC-5sv after different incubation times with AsiDNA followed or not by 10Gy irradiation. (**B**) Quantification of S2056-DNA-PK foci per nucleus in SK28 by immunostaining and microscopy analysis after various times of AsiDNA treatment and followed or not by 10 Gy irradiation. Foci were monitored 1 h after irradiation. Mann-Whitney test: ***** *p* value < 0.00001, NS: not significant.

**Figure 7 cells-11-02149-f007:**
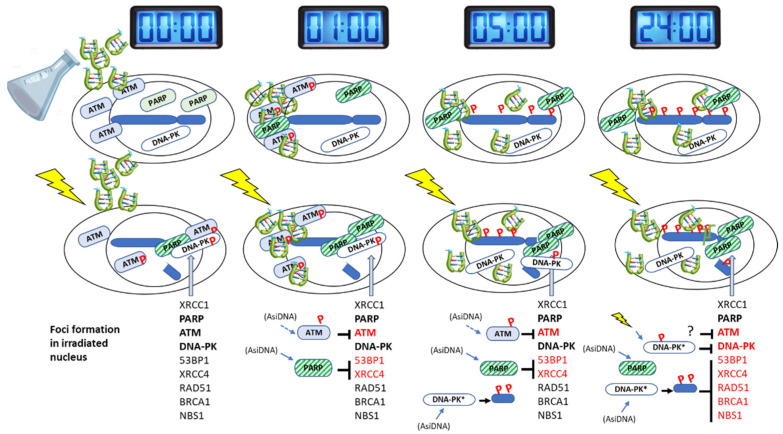
Scheme of the kinetics of events. Line 1: gray clocks indicate the length of treatment with AsiDNA. Line 2: AsiDNA molecules (green) penetrate progressively in the cells and reach nucleus within 5 h of treatment. They progressively activate target enzymes and trigger the autoparylation of PARP and phosphorylation (P) of ATM in the cytoplasm and H2AX and HSP90 in the nucleus. Line 3: differences in the phosphorylation and recruitment of repair proteins at radio-induced damage site according to the time of AsiDNA pretreatment. Line 4: list of enzymes that form foci (black) or are inhibited by AsiDNA pretreatment (red). Signaling enzymes whose activation by AsiDNA prevents focus formation are indicated in front of the inhibited proteins for repair focus formation. The three main enzymes studied here are indicated in bold, and the small arrows indicate by which these enzymes they are activated. The role of DNA-PK could be through the pan-nuclear phosphorylation of H2AX that interferes with efficient repair at double-strand breaks.

**Table 1 cells-11-02149-t001:** Inhibition of repair enzymes recruitment at DNA damage. Cells treated for different times with AsiDNA at 150 µg/mL were nonirradiated or irradiated at 2Gy for P-ATM, or at 10Gy for the other enzymes. Focus formation was analyzed by immunofluorescence at 10 min (for P-ATM) or 2 h (for the other proteins) after irradiation. All analyses were performed in SK28 and confirmed in at least one other cell line, such as MDA-MB-231, MRC-5sv, Hela, or MO59K. The quantification of foci after irradiation was performed as described in Materials and Methods. Enzyme dependence to ATM, DNA-PK, and PARP1 was estimated by focus quantification in cells treated with ATM inhibitor (KU55933), DNA-PK inhibitor (NU7026), PARP inhibitor (olaparib), or in cell lines carrying mutations or silenced for ATM, DNA-PK, or PARP1.

			AsiDNA treatment 1 h				AsiDNA treatment 5 h				AsiDNA treatment 24 h			
		Enzyme response	in DDR proficient cells	in cells inactivated for			in DDR proficient cells	in cells inactivated for			in DDR proficient cells	in cells inactivated for		
				ATM	DNA-PK	PARP1		ATM	DNA-PK	PARP1		ATM	DNA-PK	PARP1
No irradiation	ATM	Autophosphorylation	Yes	Irr	Yes	Yes	Decrease	Irr	Yes	Yes	No	Irr	Irr	Irr
	DNA-PK	Pan nuclear phosphorylation of H2AX and HSP90	No	Irr	Irr	Irr	No	Irr	Irr	Irr	Yes	Yes	No	Yes
	PARP1	PARylation activity	Yes	Yes	Yes	No	Yes	Yes	Yes	No	Yes	Yes	Yes	No
Gamma irradiation	P-ATM	Inhibition of nuclear foci	Yes	Irr	- ^(1)^	Yes	Yes	Irr	Yes	Yes	Yes	Irr	Yes	- ^(1)^
	NBS1	Inhibition of nuclear foci	- ^(1)^	Irr	Irr	Irr	No	Irr	Irr	Irr	Yes	- ^(1)^	- ^(1)^	- ^(1)^
	BRCA1	Inhibition of nuclear foci ^(3)^	- ^(1)^	Irr	Irr	Irr	No	Irr	Irr	Irr	Yes	Yes	No	Yes
	Rad51	Inhibition of nuclear foci ^(3)^	- ^(1)^	Irr	Irr	Irr	No	Irr	Irr	Irr	Yes	Yes	No	- ^(1)^
	P-DNA-PK	Inhibition of nuclear foci	- ^(1)^	Irr	Irr	Irr	No	Irr	Irr	Irr	Yes	- ^(1)^	Irr	- ^(1)^
	γ-H2AX	Pan nuclear phosphorylation	No	Irr	Irr	Irr	No	Irr	Irr	Irr	Yes	Yes	No	Yes
		Inhibition of nuclear foci? ^(4)^	No	Irr	Irr	Irr	No	Irr	Irr	Irr	No	Irr	Irr	Irr
	53BP1	Inhibition of nuclear foci	Yes	Yes	Yes	No	Yes	Yes	Yes	No	Yes	Yes	No	Yes
Laser induced damage	XRCC4	Inhibition of recruitment at damage	Yes	Yes	- ^(2)^	No	Yes	Yes	- ^(2)^	No	Yes	Yes	- ^(2)^	- ^(1)^
	PARP1	Inhibition of recruitment at damage	No	Irr	Irr	Irr	No	Irr	Irr	Irr	No	Irr	Irr	Irr
	XRCC1	Inhibition of recruitment at damage	No	Irr	Irr	Irr	No	Irr	Irr	Irr	No	Irr	Irr	Irr

^(1)^ Not tested; ^(2)^ No XRCC4 foci in DNA-PK defective cells; ^(3)^ Inhibition observed only in cells with pan nuclear *γ*H2AX; ^(4)^ Inhibition observed only in cells without pan nuclear *γ*H2AX; Irr: the tested event is not observed in the proficient cell or the enzyme response cannot be observed in this condition.

## Data Availability

Not applicable.

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
