# Peer review of "Inhibition of DNA Repair by Inappropriate Activation of ATM, PARP, and DNA-PK with the Drug Agonist AsiDNA"

_cells, 2022, doi:10.3390/cells11142149_

Round 1

Reviewer 1 Report

This manuscript, entitled "Inhibition of DNA Repair by Inappropriate Activation of ATM, PARP and DNA-PK with AsiDNA Drug Agonist", is a continuation of previous work regarding the mechanism of action and application of Dbait structures binding DNA damage signaling enzymes and thus inhibiting the recruitment at the damage site of proteins involved in DNA break repair pathways. Overall, this manuscript is well written, and the results are carefully described and discussed.  

However, there are some important issues that should be clarified before publication:

In the Abstract the authors declare that they “used stable short DNA molecules (AsiDNA) to analyse DNA repair early events from the detection of DNA breaks to the formation of DNA repair foci after irradiation”. However, I think that it is somewhat inappropriate to talk about studying the early stages of DNA repair, which occurs in the cell nucleus, using short DNA, which is mainly localized in the cytoplasm. The authors consider these AsiDNA as potential therapeutic inhibitors of the DNA break repair, so it would be better to talk about studying the mechanism of the repair inhibition, rather than the mechanism of repair itself.

In the Introduction section, the authors discuss the recruitment and activation kinetics of ATM, DNA-PKcs, and PARP1 and write that the choice of repair pathway “could be determined by the recruitment of the first proteins binding the DNA end”. However, this is not entirely correct, since, first of all, the choice of the repair path depends on the stage of the cell life cycle. NHEJ can occur at any stage, while HR requires an intact sister chromatid and can only occur in S and G2 phases.

The results presented in this article show that DNA-PK activation takes several hours after the cell treatment with Dbait. However, in the previous article of this group (doi: 10.1074/jbc.M111.320887), DNA-PK activation occurs within one hour. This inconsistency of results needs to be discussed.

Figure 3A shows the localization of P-ATM in cells treated with AsiDNA at different times and irradiated or not. In the case of irradiated cells, foci with colocalization of P-ATM and γH2AX are visible at time 0h. However, at 0.5-5 hours, an increase in P-ATM and γH2AX in the cytoplasm and a decrease in the nucleus can be seen. Does this mean that, in the presence of AsiDNA, P-ATM (or ATM) migrates from the nucleus to the cytoplasm?

In the section 3.4 it is noted that HSP90 is specifically phosphorylated by DNA-PK. However, it is not correct as there are works showing that it can also be phosphorylated by ATM. Moreover, the manuscript of Elaimy AL. et al. states that ATM is the primary kinase responsible for phosphorylation of Hsp90α after ionizing radiation (doi: 10.18632/oncotarget.12557). This should be taken in consideration and discussed.

Reviewer 2 Report

SUMMARY

The authors try to unpick the cellular signalling responses to treatment with AsiDNA molecules, and then go on to further characterise the response to radiation at various time points following the AsiDNA treatment. The results are of significant interest but further work is required, both in the form of additional experiments and a correction of errors in the text, before publication.

GENERAL COMMENTS

There are significant language mistakes throughout the paper, both grammatical and incorrect word usage, that must be corrected. Specific examples include:

Line 36-39: “To protect cells against such events, several repair pathways exist which are complementary and operate on different end substrates: namely the Non Homologous End Joining (NHEJ), its alternative form PARP dependent (alt-NHEJ) and the Homologous Recombination (HR). All involve complex cascade of protein recruitment at the damage site.” Would be clearer as: “To protect cells against such events, several complementary repair pathways exist: Non Homologous End Joining (NHEJ), its alternative PARP dependent form (alt-NHEJ), and Homologous Recombination (HR). All operating on different end substrates and involving a complex cascade of protein recruitment at the damage site.”

Line 183: biotin, not biotine

Line 210: either, not even

Line 225: and, not et

However, given the large number I would suggest getting someone to specifically edit the manuscript for these kinds of errors rather than leaving it to the reviewers to pick all of them up.

It is not described exactly what the authors mean by AsiDNA treatment for all of the experiments. It is described is the legend of figure 1A and the legend for table 1, is this the same for all experiments? This should be made clear.

Experiments are presented with incomplete sets of timepoints with respect to each other. The reasoning for this is unclear and greatly weakens the impression provided by the data. The work detailing the PARP and ATM responses seems robust, if the text and figures are adjusted, but the DNA-PK response requires further experiments detailed below. Overall, I believe the theory put forward by the authors may well be right, but unfortunately the data here is not yet sufficiently strong to support this conclusion.

SPECIFIC COMMENTS

Results section 3.1

Line 159: Figure 1A shows convincing evidence for the uptake of the AsiDNA molecules, however the quantification in figure 1B has error bars that are so large they bring into question whether the results are significant at the different time points. More cells should be counted at the 5 hour timepoint and as other experiments are continued for 120 hours it would also be beneficial to include data at these longer timepoints to demonstrate whether the AsiDNA is still present for the duration of the experiments (where PARP signalling has been demonstrated to be active).

Line 174: The smooth graphs presented in figure 1C appear to have features that do not seem to correspond to the timepoints shown in the supplementary data? The points at which measurements were made should be clearly shown on this graph alongside a description of how it was produced.

Line 177: If the authors wish to make the claim that the kinetics are unaffected by the other pathways they need to include data for all the timepoints used in the study in figure 2 (or an additional supplementary figure), not just a single one (5hours for ATMi, 24 hours for PARPi and unspecified for DNAPKi) for each measure. While the representative images for pATM and yH2AX appear to support the claims, statistical quantification of this data should also be provided for figures 2A and 2C.

See section 3.4 for comments about DNAPK activity.

Results section 3.2

Line 208: I do not understand. The data clearly show cytoplasmic pATM at whatever the 0.5-5 hour timepoint represents, so how this resumes at the 10 hour mark is unclear. Do the authors mean nuclear? And if so, data demonstrating this should be provided in figure 3.

How long after irradiation are the images taken for pATM and NBS1? The materials and methods suggest ten minutes and two hours respectively. Why? If additional time points post irradiation are taken are the results the same? Expanding the analysis of foci formation following irradiation to more timepoints post irradiation would strengthen the analysis.

The use of the label 0.5-5hours in figure 3A is not appropriate, the image clearly represents a static timepoint, what is it? The inclusion of images at multiple time points from 30 minutes to 24 hours, and the associated quantitative analysis, would greatly benefit this figure.

All timepoints for the experiments should be quantified in figure 3B.

Given the inclusion for everything else a representative image for NBS1 foci should be included for figure 3C.

Results section 3.3

The results detailed in figure 4 are clear. They would benefit from the inclusion of longer timepoints for figure 1B though to understand if the AsiDNA is still present in the cells at the 120 hour timepoint where PARP activation is still observed. Line 244 requires a reference for the PARP-XRCC1 interaction.

Results section 3.4

This section is problematic, significantly weakens the paper, with issues that must be addressed.

Using yH2AX as a marker for DNAPK activity is not ideal as the mark can also be put on by ATM and ATR. The experiment would need to be repeated with these enzymes inactivated if the authors wish to use this mark (as in figure 2 for ATMi but for all timepoints and also using ATRi). HSP90 pT7 is also not specific to DNA-PK, and given the reference provided for this is the authors own paper, it is somewhat surprising they are not aware of this.

A better option would be to choose a mark that is specific for DNAPK such as its autophosphorylation on Serine 2056, but the authors show later in the paper that this mark isn’t found following AsiDNA treatment. This is surprising, if DNAPK is indeed active, and requires further investigation. I suggest repeating this analysis, and that of section 3.4 using another marker of DNA-PK activity, such as phosphorylation of the Ku heterodimer for which commercially available antibodies can be found.

An additional time course including DNAPKi alongside AsiDNA treatment should be provided in figure S2 to help demonstrate the activity is DNAPK dependent.

Line 271: I do not agree that there is no cytoplasmic pHSP90 signal in the 24hr timepoint in figure S3.

Line 278: The image provided is not completely clear but comparing to the 24 hr AsiDNA picture, I do not agree that there are no yH2AX foci alongside the pan nuclear staining for the 24 hour AsiDNA + 10Gy image.

As such the claims of this section are unsubstantiated or at odds to the provided data.

Results section 3.5

Line 290: There is at least one further timepoint at 120 hours, giving five in total, where PARP is still active but DNA-PK and ATM are not. Why was this not included in the analysis? According to the theory put forward PARP signalling should still be up but with few ill effects for the recruitment of repair components following the reduction of the DNA-PK activity at this time point. Including this would improve the analysis.

Line 297: In figure S5A there are 53BP1 foci at the five hour timepoint in contradiction of the text. Would suggest weakening the statement to ‘foci numbers were reduced’. In figure 3A and B no data for the five hour timepoint for ATM foci is provided so the statement is unsubstantiated.

Line 298: In figure S6 it would be good to include the quantification for NBS1 as well.

Line 300: again this statement is too strong for the evidence, there clearly are foci, they are fewer, or weaker, and this is a significant difference, but they are there.

Line 302: it is unclear what the authors mean by this or what evidence they are using to support it.

Line 305: PARP is not essential for XRCC4 recruitment and reference 25, provided to support this statement, makes no such claim, rather it states it is required for expansion of XRCC4 foci therefore the analysis about this is incorrect.

Table 1: there are a lot of ‘not tested’ time points especially for the 5hr AsiDNA treatment with irradiation. Experiments should be repeated so that these are tested.

Round 2

Reviewer 2 Report

Point 1 - Okay.

Point 2 - Okay.

Point 3 - It isn’t good enough to say that. “How these molecules are protected in specific locations or complexes and do not follow the rapid decrease observed on microscopy remain to be elucidated.”. If the authors’ theory is that the molecules are present and causing signalling they should be observable.

Point 4 - Okay. There still seems to be timepoints for which there is not supplementary data (i.e. ATM at ~15 and 20 hours, DNA-PK at the points past 24 hours) but I guess they are just not included due to space constraints?

Point 5 - I don’t accept this. The analysis needs to be done at different timepoints to justify the claim.

Point 6 - Okay.

Point 7 - Okay, I accept these are standard timepoints but given the nature of this study is about changes over time I still believe other timepoints must be done.

Point 8 - Okay.

Point 9 - This doesn’t make sense. Figure S1B is just with AsiDNA treatment, this figure is also for AsiDNA treatment and 2Gy radiation which is still not quantified?

Point 10 - Okay.

Point 11 - Okay. I accept the authors are no longer able to carry out the EM experiment and have attempted another method instead. However, the concentration is given in uM here but ug/ml previously, I am therefore not sure if this represents a massively increased dose? Given this experiment has now been done the data should be included in the manuscript.

Point 12 - I maintain that the analysis including ATMi and ATRi must be done for all time points.

Point 13 - Yes, this is my point, it is unusual to have active DNA-PK that is not autophosphorylated. I am not sure why the authors think the presence of AsiDNA is lower than that of a double strand break if there is sufficient activity for pan nuclear yH2AX production? Given this is the major finding it should be investigated further.

Point 14 - Figure 2B shows PAR data. Figure 2C shows a single image for a DNAPK-/- cell line. I still think the experiment using an inhibitor for DNA-PK over 120 hours must be done to allow conclusions to be derived for the complete time course.

Point 15 - Okay.

Point 16/17 - Okay, but there do appear to be yH2AX foci alongside the pan nuclear staining which is at odds with this statement (bottom right panel of figure S4).

Point 18 - I agree with the authors prediction... they should test it.

Point 19 - Okay.

Point 20 - Okay.

Point 21 - Okay.

Point 22 - Okay.

Point 23 - Okay.

Round 3

Reviewer 2 Report

Point 3:

There is now a disconnect between the measurement by the two methods, with one suggesting cells are positive at all timepoints, with the signalling dropping ~3.5 fold in the EM and ~1.2 fold for the flow cytometry from the 5 to the 24 hour timepoints. If the authors suggestion that the loss of biotin was preventing the detection by EM but the molecule is still there to explain this discrepancy with the Cy-5 data then this doesn’t sit well with the rest of the paper where distinct differences are seen for the different timepoints. If the molecule remains in the cells unaltered throughout the timepoints measured why is there a difference in what the cell is doing in response to it?

Point 4:

Okay.

Point 5:

Excellent.

Point 13:

Okay. I accept the context is different and the inhibitor data for yH2AX and HSP90 is compelling. However my understanding is that DNA PK activation requires both DNA and Ku binding and that once activated it phosphorylates itself and Ku. I still don’t understand why this phosphorylation of DNAPK isn’t seen here in the absence of further DNA damage.

If the authors are using the presence of the cholesterol as a defence, are the authors not also concerned that if the ends of the DNA are not available then they are not going to be able to interact with DNAPK and Ku?

Point 14:

Okay

Point 16/17:

I maintain that in that figure panel, in the pan nuclear stained cells, there are foci. This is a problem. I’m not sure what more there is to say.

Point 18:

Fair enough, this might be tending to completionism on my part, I accept that while interesting this is beyond the scope of what the authors are trying to show.
